

# Benefits of sub-component over full-scale blade testing elaborated on a trailing edge bond line design validation

Malo Rosemeier[1], Gregor Basters[1], and Alexandros Antoniou[1]

[1]Divison Structural Components, Fraunhofer Institute for Wind Energy and Energy System Technology IWES, Am Seedeich 45, 27572 Bremerhaven, Germany

*Correspondence to:* Malo Rosemeier (malo.rosemeier@iwes.fraunhofer.de)

**Abstract.** Wind turbine rotor blades are designed and certified according to the current IEC (2012) and DNV GL AS (2015) standards, which include the final full-scale experiment. The experiment is used to validate the assumptions made in the design models. In this work the drawbacks of traditional static and fatigue full-scale testing are elaborated, i. e. the replication of realistic loading and structural response. Sub-component testing is proposed as a potential method to mitigate some of the drawbacks. Compared to the actual loading that a rotor blade is subjected to under field conditions, the full-scale test loading is subjected to the following simplifications and constraints: First, the full-scale fatigue test is conducted as a cyclic test, where the load time series obtained from aero-servo-elastic simulations are simplified to a damage equivalent load range. Second, the load directions are typically applied solely in two directions, often pure lead-lag and flap-wise directions which are not necessarily the most critical load directions for a particular blade segment. Third, parts of the blade are overloaded by up to 20% to achieve the target load along the whole span. Fourth, parts of the blade are not tested due to load introduction via load frames. Finally, another downside of a state-of-the-art, uni-axial, resonant, full-scale testing method is that dynamic testing at the eigenfrequencies of today's blades in respect of the first flap-wise mode between 0.4Hz and 1.0Hz results in long test times. Testing usually takes several months. In contrast, the sub-component fatigue testing time can be substantially faster than the full-scale blade test since (a) the load can be introduced with higher frequencies which are not constrained by the blade's eigenfrequency, and (b) the stress ratio between the minimum and the maximum stress exposure to which the structure is subjected can be increased to higher, more realistic values. Furthermore, sub-component testing could increase the structural reliability by focusing on the critical areas and replicating the design loads more accurately in the most critical directions. In this work, the comparison of the two testing methods is elaborated by way of example on a trailing edge bond line design.

## 1 Introduction

Reliability, i. e. serviceability and structural integrity of rotor blades, is essential to fulfill the requirements placed on a wind turbine in the field. Serviceability is of economic interest while integrity is of interest from a safety point of view. Before a rotor blade design goes into operation, a full-scale blade test (FST) is required in the certification process to validate the assumptions made in the design models. Wind turbine rotor blades are designed and certified according to the current standards and guidelines (IEC (2012), DNV GL AS (2015)). As part of the certification process, static as well as fatigue loads are applied



to the rotor blade. Static and fatigue loads are usually applied in the two main directions, i. e. lead-lag and flap-wise. Damage equivalent fatigue test loads are applied to the rotor blade, but they do not necessarily reflect the actual load direction, amplitude and mean during service life.

Although DNV GL AS (2015) accepts combined static full-scale tests (SFST) rather than pure lead-lag and flap-wise, whose importance is highlighted in Roczek-Sieradzan et al. (2011) and Haselbach et al. (2016), this approach is rarely used for certification tests.

Attempts have been made to improve fatigue full-scale blade testing (FFST) towards a more realistic scenario by minimizing the overloading, i. e. the deviation between applied test bending moments and target bending moments as per the design requirements. For example, Lee and Park (2016) have applied an algorithm to optimize the bending moment distribution for uni-axial, resonant FFST by using additional masses, and determining optimal actuator positions and excitation frequencies. Bi-axial testing is yet another approach to emulate a multi-axial stress state within the areas of interest. It comprises alternative excitation techniques in FFST and has been a focus of research for a number of years (White, 2004; White et al., 2005; Greaves et al., 2012; Eder et al., 2017)). All FFST approaches, however, focus on minimizing the overloading but still fall short when it comes to emulating the loading as it would occur in the field.

IEC (2012) states that during fatigue testing, the mean loads applied shall normally be as close as possible to the mean load under the operating conditions that cause the most severe fatigue damage. The importance of considering mean loading for the fatigue life evaluation of fiber reinforced polymers (FRP) is described and quantified in Sutherland and Mandell (2005).

According to Krimmer et al. (2016) lead-lag fatigue loading is the design-driving loading for current and future rotor blades. Therefore, this work focuses on the most sensitive area for the lead-lag load case, i. e. on the trailing edge bond line in particular.

In the past, generic beam elements have been tested to investigate the behavior of adhesive joints under static and fatigue loading (Sayer et al., 2012). Branner et al. (2016), Rosemeier et al. (2016) and Lahuerta et al. (2017) have proposed sub-component tests (SCT) to investigate the structural performance of trailing edge bond lines in particular.

Within this work, SCT is presented as a means to potentially overcome some of the drawbacks of FST. By taking the target mean loading, and thus the stress ratio between the minimum and the maximum stress exposure at the trailing edge into account, the structural evaluation of the blade can be approached more realistically in field simulations by conducting SCT rather than FST. Furthermore, testing time can be reduced significantly if only one sub-component of interest is considered.

First, FST and SCT concepts are described. Second, the benefits of SCT over static and fatigue FST are highlighted on the basis of calculations.

## 2 Full-scale blade and sub-component testing concepts

For a static full-scale blade test (SFST), the blade is clamped to a stiff block and several load frames are attached to the blade along the span (Fig. 1a). Point loads are introduced via load frames pulling ropes which are connected to actuators on a strong floor.





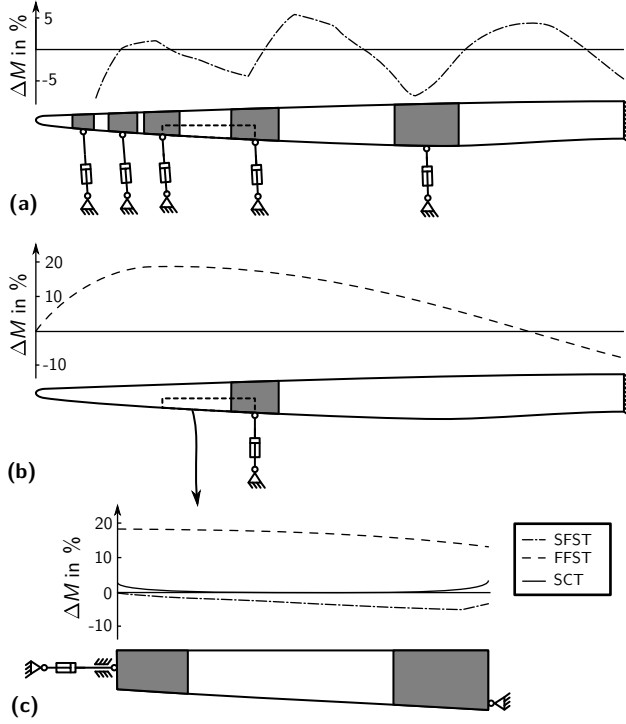

**Figure 1.** Load deviation $\Delta M$ between test bending moment and target bending moment determined for the DTU10MW blade and test setup for static full-scale test (SFST) in side view (excerpt from Antoniou et al. (2015)) (a), fatigue full-scale test (FFST) in top view (b), and blade cut-out sub-component test (SCT) (c). Untested areas are shown in gray.

For a uni-axial, resonant, fatigue full-scale blade test (FFST), only one load frame is attached to the blade. Cyclic tension and compression loads are applied by one actuator (Fig. 1b). The testing eigenfrequency is chosen so as to be close to the blade's eigenfrequency, thus enabling displacement-controlled resonant excitation of the mode shape of the blade.

The concept of blade sub-component testing (SCT) followed in this work is based on Rosemeier et al. (2016). Here, the blade
5 segment of interest can be either cut out or manufactured separately. The segment is further cut in the span-wise direction such that only the area of interest remains, in this case the trailing edge cell including one main shear web (Fig. 1c). This is done to obtain a closed cross-section, which can still realistically emulate "breathing" or "pumping" effects of the thin-walled blade structure. Both ends of the sub-component are glued into load frames, which are connected to ball joints. An eccentric axial load is introduced at one joint, thus introducing a normal force on which a bending moment is superposed. The position of the
10 two joints within the cross-sectional plane can be chosen arbitrarily, which makes it possible to introduce any load combination and distribution of lead-lag and flap-wise loading.



## 3  Comparison of sub-component with full-scale blade testing

In the following, the benefits and drawbacks of sub-component testing (SCT) over static and fatigue full-scale blade testing (FST) are elaborated.

An advantage of SCT over FST is that the number of relatively inexpensive specimen can be increased to investigate different
design variants of a critical blade detail, for example. Owing to the smaller dimension and relatively small displacements of an SCT, it is possible to conduct experiments under environmental conditions, e. g. in climate chambers, and to better observe the experiment via optical measurement systems, such as digital image correlation, for example.

A disadvantage of SCT is that only a segment of the blade is considered for testing. Furthermore, the design of the boundary conditions of an SCT relies on detailed models, e. g. for the determination of blade properties of a cut-out segment as explained
in Rosemeier et al. (2016).

The focus of the following sections is on the problem of constrained areas and overloading of blade parts, the load direction in static and fatigue testing, as well as on the stress ratio at the trailing edge and the resulting testing time.

### 3.1  Constrained areas and overloading

Load frames diminish tested areas because they constrain the structure (Fig. 1a). The longitudinal dimension of the constrained
area at a load clamp is assumed to be $\pm 75\%$ of the local blade chord length (IEC, 2012). Critical cross-sections should not be within constrained areas. Owing to the multi-linear moment distribution in the static test, the critical areas are necessarily in under- or over-loaded areas (Fig. 1a). Thus, due to the setup of an FST, these areas are subjected to a load deviation, which means a lower or a higher test loading $M_{\text{test}}$ compared to the target load envelope $M_{\text{target}}$:

$$\Delta M = \frac{M_{\text{test}}}{M_{\text{target}}}. \tag{1}$$

In the fatigue FST, the constrained area is reduced to solely one load clamp, which increases the tested area for this test (Fig. 1b). The load deviation $\Delta M$, however, can be even higher for the blade (overloading by up to $20\%$) although mass tuning is conducted. Overloading occurs along all tested areas because the shape of the target moment distribution cannot be fully replicated by the test moment distribution. That is, parts of the blade can be damaged far beyond their fatigue life before all relevant areas reach their target damage.

In an SCT, the constrained areas are determined by means of a method similar to that used in FST (Fig. 1c). This implies that a reasonable length of the specimen should be chosen to replicate, for example, a realistic buckling response. The overloading in the tested areas, however, is close to zero, since the bending moment can be adjusted by specially designed boundary conditions (Rosemeier et al., 2016).

Moreover, in an SCT the cross-section of interest can be tested up to its final fatigue limit state compared to an FST, where
other, highly overloaded blade areas may limit the damaging that is required to reach the final fatigue limit state at the cross-section of interest.





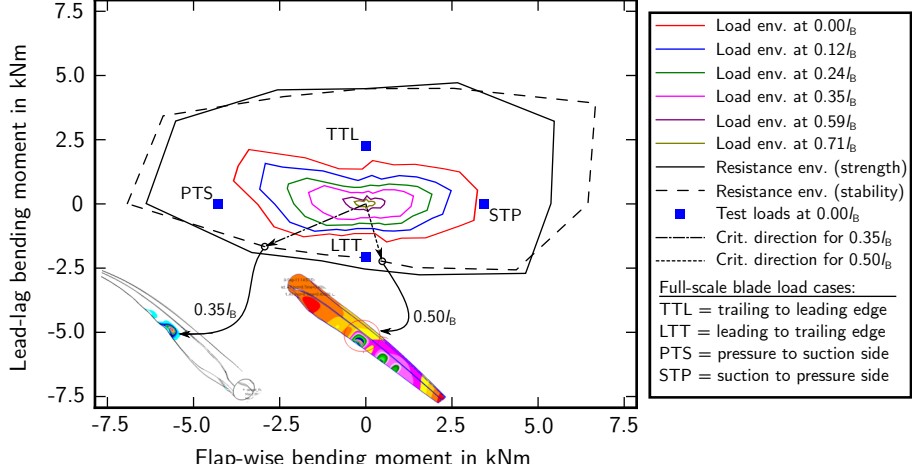

**Figure 2.** Load envelopes in terms of moment vectors determined from aero-servo-elastic simulations for several cross-sections of a 34 m blade and certification test loads contrasted with strength and stability resistance of the whole blade in terms of the root bending moment (excerpt from Branner et al. (2015)). The two moment vectors represent the critical load direction at 35% blade length ($l_B$) and 50% $l_B$.

## 3.2 Load direction in static testing

When designing a static FST, the usual procedure is to use a "worst" case load envelope of each cross-section along the blade span (Fig. 2) to create the target loads. From these envelopes at least two load directions are extracted for the static load case. Either the pure lead-lag or flap-wise load cases (PTS, STP, LTT, and TTL as explained in Fig. 2) or any combination

of these load directions can be chosen for the tests. Considering the stress exposure (3) of the critical area of interest of each cross-section individually, however, the load direction leading to the critical stress exposure is not necessarily identical with the overall full-scale loading directions.

     In Fig. 2 the load envelopes at different cross-sections along the blade are plotted against the strength and stability resistance envelope of all cross-sections, i. e. the whole blade. Furthermore, the load direction with the critical stress exposure is

highlighted by way of example for two different cross-sections at 35% blade length ($l_B$) and 50% $l_B$, where the critical load directions are a combination of PTS plus LTT, and STP plus LTT, respectively.

     Theoretically, the load envelopes should be compared to the resistance envelopes at each particular cross-section to determine the most critical load directions. The determination of the resistances for each cross-section is only possible via analysis of a blade segment or sub-component cut-out from the full-blade subjected to the load envelope determined from aero-servo-elastic

simulations.

     SCT allows for the flexible adjustment of these critical loading configurations for particular blade segments on an individual basis.





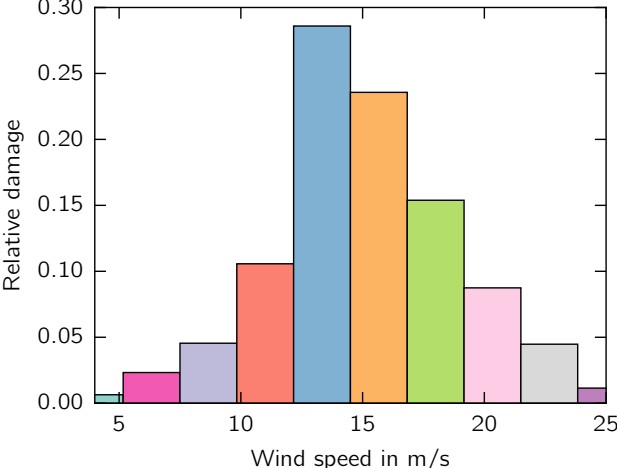

**Figure 3.** Relative damage of different wind speed bins at 10 % blade length.

## 3.3 Load direction in fatigue testing

Considering the load direction, the uni-axial, resonant FFST is constrained to the static blade dead load (pressure-side points towards the strong floor) superposed with the dynamic inertia loads of the mode shape of the blade. According to IEC (2012), the mean loads applied during FFST shall be as close as possible to the mean load at the operating conditions that cause the most severe fatigue damage.

Since no load directions other than the first and second modal shapes, i. e. lead-lag and flap, are possible, the only degree of freedom during uni-axial, resonant FST is the direction of the gravity load due to the pitching position of the blade.

The loads for FFST and field calculations are elaborated by way of example using the DTU10MW reference turbine (Bak et al., 2013) in respect of the loading of the trailing edge bond line. A simplified load calculation was conducted comprising the superposition of the quasi-static mean flap-wise aerodynamic load for ten wind speed bins plus the lead-lag gravity load due to the rotor revolution as a function of the blade pitch angle.

A beam model implemented in ANSYS APDL (Swanson, 2014) was assembled with a fully-populated cross-section stiffness matrix determined by the Beam Cross Section Analysis Software BECAS (Blasques and Stolpe, 2012; Blasques, 2014; Blasques and Bitsche, 2015; Blasques et al., 2015). The blade parametrization and input generation was conducted using workflows of the FUSED-Wind framework (Zahle et al., 2015). The BEM-based aerodynamic rotor simulator CCBlade (Ning, 2014) was used and populated with airfoil polars determined by Rfoil, an extension of Xfoil (Drela, 1989) including rotational effects (Bosschers, 1996).

The lead-lag load cycles were determined on the basis of wind speed distribution and the rotor revolution as well as the design life of 20 years. Furthermore, damage equivalent loads were determined following Freebury and Musial (2000) using an S-N exponent of $m = 10$ and ultimate loads as of Bak et al. (2013).

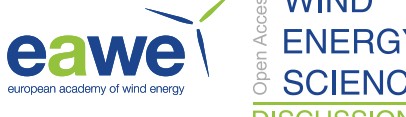



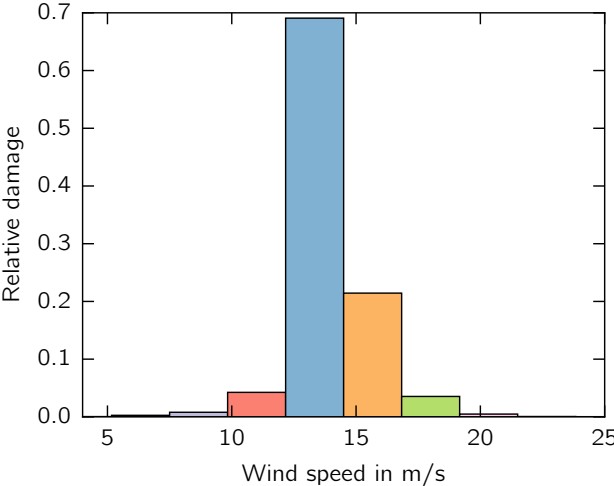

**Figure 4.** Relative damage of different wind speed bins at 70 % blade length.

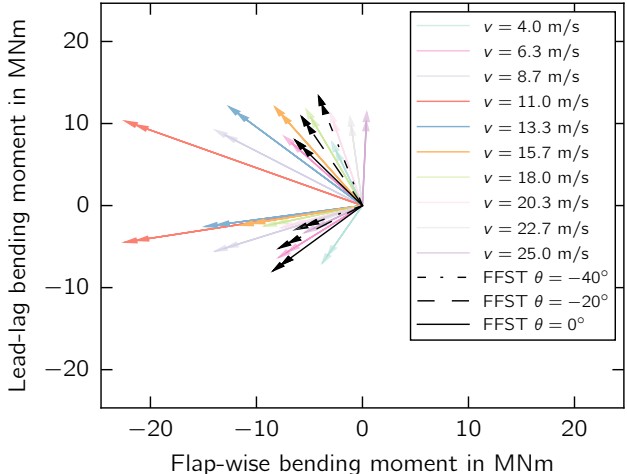

**Figure 5.** Moment vectors of lead-lag loading at different wind speed bins and in a lead-lag fatigue full-scale blade test (FFST) at different pitch angles at 10 % blade length.

The greatest impact on the damage was found at wind speeds around rated conditions between 11.0m/s and 15.6m/s (Fig. 3 and 4). Berring et al. (2014) have found similar results for a 34m blade. The relative damage impact peak at around rated wind speed of an outboard cross-section (Fig. 4) protrudes over the peak of an inboard cross-section (Fig. 3) because in outboard regions the influence of aerodynamic loads predominates.

5    Furthermore, a uni-axial, resonant, lead-lag FFST was designed. To this end, the lead-lag target loads were calculated on the basis of damage equivalent loads and scaled according to Palmgren (1924) and Miner (1945) to a cycle number of $3.0\mathrm{E}+06$ using an S-N exponent of $m = 10$. The test loads were determined on the basis of the inertia loads caused by the dynamic



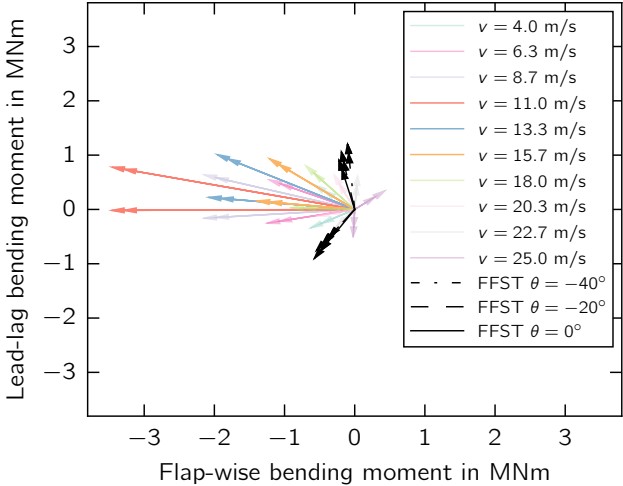

**Figure 6.** Moment vectors of lead-lag loading at different wind speed bins and in a lead-lag fatigue full-scale blade test (FFST) at different pitch angles at 70 % blade length.

excitation of the first lead-lag dominated eigenmode extracted from a modal analysis. This load distribution was superposed with the gravity loading according to the FFST setup for different pitch angles $\theta$, where $\theta = 0°$ corresponds to the suction side pointing towards the strong floor, and a negative pitch angle which rotates the leading edge towards the floor.

The minimum and maximum bending moment vectors acting at each wind speed bin according to field calculations and for three pitched FFSTs are shown for two cross-sections at 10% and 70% blade length (Fig. 5 and Fig. 6). From the angle between minimum and maximum moment vector at the inboard cross-section it can be seen that the lead-lag gravity loads dominate.

Moreover, Eder and Bitsche (2015) have shown that the quantity of geometrically non-linear deformations, e. g. "breathing" or "pumping" of the trailing edge panel, depends on the load direction. In particular, this deformation is expected to be most prominent for moment vectors within the second and fourth quadrants of the cross-section coordinate system as shown in Fig. 5 and Fig. 6. Pitching the blade in an FFST helps to adjust the mean load direction towards field load directions for one cross-section along the span, while in an SCT setup any of the loading scenarios for the different wind speed bins which are shown can theoretically be replicated, leading to a more realistic loading condition in the test.

Using the loading of the most severe wind speeds ($11.0\mathrm{m/s}$, $13.3\mathrm{m/s}$, $15.7\mathrm{m/s}$), the longitudinal strains in the direction of the blade span at the trailing edge bond line were determined for the minimum and maximum amplitudes (Fig. 7). It can be seen that the trailing edge bond line is loaded more in compression during an FFST at $\theta = 0°$ compared to the strains determined from field load calculations.

Assuming linear elastic material behavior, the stress ratio $R$ can be expressed as strain ratio between the minimum and maximum longitudinal strains:

$$R = \frac{\varepsilon_{\mathrm{lmin}}}{\varepsilon_{\mathrm{lmax}}}. \tag{2}$$





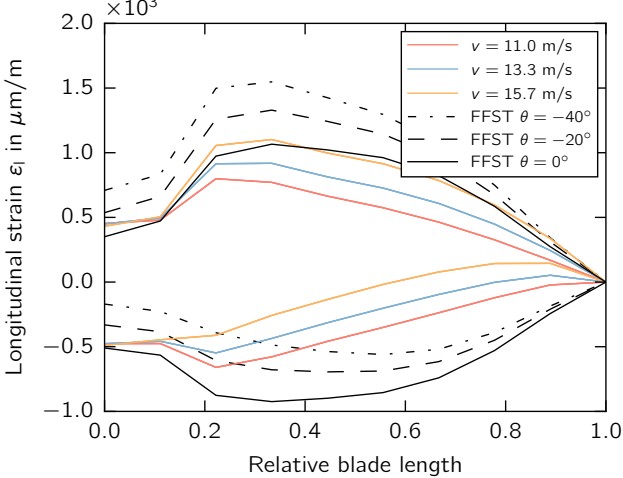

**Figure 7.** Axial strain along the trailing edge bond line at different wind speed bins, and lead-lag fatigue full-scale blade tests (FFST) at different pitch angles.

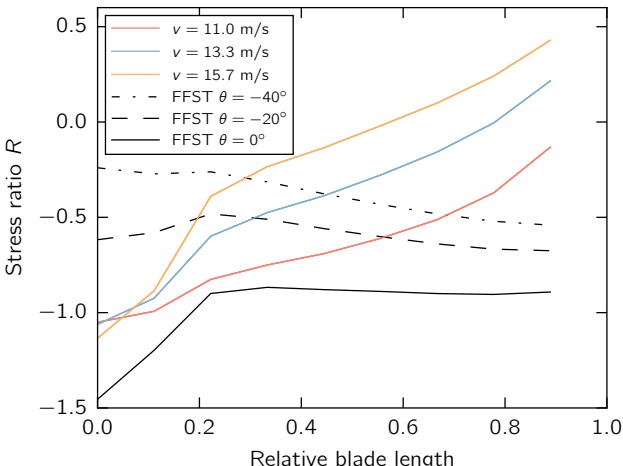

**Figure 8.** Stress ratio $R$ of the axial strain along the trailing edge bond line at different wind speed bins, and lead-lag fatigue full-scale blade tests (FFST) at different pitch angles.

Considering the $R$ of the field load calculation (Fig. 8), the deviation becomes even more prominent towards the tip, where the loading tends more towards a tension-tension rather than a tension-compression loading condition. Pitching the blade in an FFST increases the stress ratio, but the stress ratio distribution is opposite to that obtained in field simulations. Thus, in an FFST only the stress ratio of a defined cross-section can be obtained via pitching. As already stated above, in an SCT the stress ratio can be adjusted according to the cross-section of interest.



### 3.4 Stress ratio and testing time

The testing time is compared for the two testing concepts FFST and SCT. To this end, the loading at the trailing edge bond line due to FFST is compared to field calculations.

According to Krimmer et al. (2016), the internal loading of a material is expressed in terms of stress exposure $e$, which means an ambient stress $\sigma$ over the allowable stress (fracture resistance), which is here the tensile strength $R^t$. Thus the stress exposure is generally defined as

$$e = \frac{\sigma}{R^t}. \tag{3}$$

Furthermore, the materials of a rotor blade, i. e. adhesive, resin and fiber, can be considered to be isotropic. According to the symmetric constant life diagram of an isotropic material, the allowable cycle number to failure $N_i$ for a given load collective $i$, with a mean stress exposure $e_{im} = \frac{|\sigma_{im}|}{R^t}$ and a stress exposure amplitude $e_{ia} = \frac{|\sigma_{ia}|}{R^t}$, is derived from:

$$N_i = \left(\frac{1 - e_{im}}{e_{ia}}\right)^m \tag{4}$$

This means that the cycle number is directly related to the mean stress exposure, the stress exposure amplitude, and to the material-dependent S-N curve exponent $m$.

Assuming the same S-N curve exponent $m$ for different stress ratios

$$R = \frac{e_{imin}}{e_{imax}} = \frac{e_{im} - e_{ia}}{e_{im} + e_{ia}}, \tag{5}$$

the allowable load cycle number of a load collective of any ratio $N_{iR}$ and the allowable load cycles $N_{iR=-1}$ can be expressed as the relation (Appendix A):

$$\frac{N_{iR}}{N_{iR=-1}} = \left(1 - e_{ia}\frac{1 + R}{1 - R}\right)^m. \tag{6}$$

The relation is plotted for different stress exposure amplitudes in Fig. 9.

The pure testing time in days can be determined using

$$T_{test} = \frac{N_{test}}{f_{test}} \cdot \frac{1d}{86400s}, \tag{7}$$

where $N_{test}$ corresponds to the number of test cycles and $f_{test}$ to the test frequency.

For a lead-lag FFST of the DTU10MW blade in a pitching position of $\theta = 0°$ and a test frequency equal to the blade's eigenfrequency of $f_{test} = 0.965\text{Hz}$, a stress ratio of $R = -0.9$ with an stress exposure amplitude of $e_a = 0.25$ and $N_{test} = 3.0\text{E}+06$ results in a testing time of $T_{test} \approx 36\text{d}$. If the blade is pitched at $\theta = -20° \ldots -40°$, the stress ratio is increased from $R = -0.9$ to $R = -0.5$ and gets closer to stress ratios from field load calculations (Fig. 8). To overcome too high tension strains (Fig. 7), the stress exposure amplitude is decreased to $e_a = 0.125$. Owing to the effect of a higher stress ratio the testing time can be reduced by 23% to $T_{test} \approx 28\text{d}$ (green dot vs. red dot in Fig. 9).





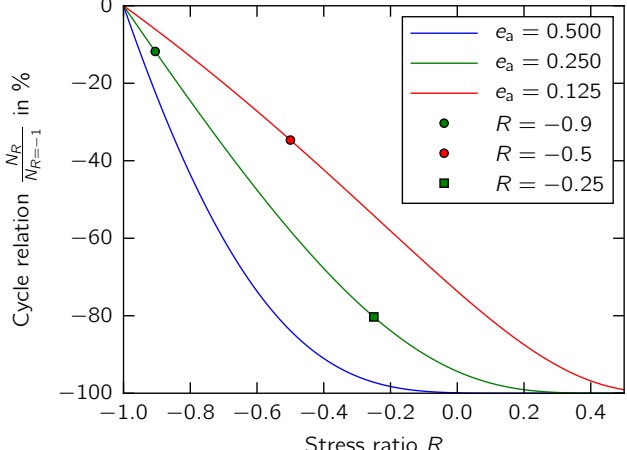

**Figure 9.** Impact of stress ratio on the relation between allowable cycles using any testing stress ratio and a stress ratio of $R = -1$ (6) shown for different stress exposure amplitudes. An S-N curve exponent of $m = 10$ was used.

When performing an SCT of a blade segment at $10\%$ blade length, the stress ratio of $R = -0.9$ is close to field calculations (Fig. 8). For this case, the testing time cannot be reduced by the testing cycles $N_{\text{test}}$ but only by an increase in testing frequency. Assuming a frequency of $f_{\text{test}} = 1.0\text{Hz}\dots1.5\text{Hz}$, the testing time is slightly to moderately reduced to $T_{\text{test}} \approx 35\text{d}\dots23\text{d}$ compared to an FFST at $\theta = 0°$. When a segment at $70\%$ blade length is considered for an SCT with a realistic stress ratio of

$R = -0.25$ from field calculations (Fig. 8) and an stress exposure amplitude of $e_{\text{a}} = 0.25$, the test cycles can be further reduced by $47\%\dots68\%$ compared to an FFST with a pitch angle between $\theta = -20°\dots-40°$ (green square vs. red dot in Fig. 9). Assuming a test frequency between $f_{\text{test}} = 1.0\text{Hz}\dots1.5\text{Hz}$, the testing time results in $T_{\text{test}} \approx 11\text{d}\dots7\text{d}$.

## 4   Conclusions

The loading conditions of a combined static (SFST) and a lead-lag fatigue full-scale blade test (FFST) were compared to field
simulations.

It was demonstrated that SFST does not necessarily cover all critical loading conditions along the blade length. Blade sub-component testing (SCT) does allow the flexible adjustment of the load direction for each segment of interest individually, however. Thus, the use of SCT could increase the structural reliability by covering all relevant loading directions compared to testing the blade solely in its full-scale loading directions.

An SCT, however, considers only one blade segment. Furthermore, the design of the boundary conditions of an SCT relies on detailed models.

Moreover, it was calculated that the load directions and the stress ratios at the trailing edge bond line differ significantly in part between field and FFST. Pitching the blade in an FFST setup can affect the load directions and ratio for a particular cross-section, but does not necessarily replicate the load directions and ratios along the blade span under field conditions. SCT


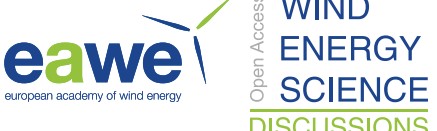

has the potential to overcome these drawbacks while replicating different loading conditions more realistically than FFST. Furthermore, using SCT means the testing time of one blade segment can be significantly reduced when the field stress ratio is larger than the stress ratio the blade is subjected to in FFST.

## Appendix A: Derivation of Equation 6

5   After several rearrangement steps, (5) can be written as:

$$e_{im} = e_{ia}\frac{1+R}{1-R}. \tag{A1}$$

Furthermore, the initial steps deriving (6) are:

$$\frac{N_{iR}}{N_{iR=-1.0}} = \frac{\left(\frac{1-e_{im}}{e_{ia}}\right)^m}{\left(\frac{1}{e_{ia}}\right)^m} = (1-e_{im})^m. \tag{A2}$$

(A1) and (A2) can be used to derive (6).

10   *Acknowledgements.* We acknowledge the support of the European Commission's Seventh Framework Programme within the IRPWind project (609795) and the support within the Future Concept Fatigue Strength of Rotor Blades project granted by the German Federal Ministry for Economic Affairs and Energy (BMWi) (0325939) and the Senator for Health, Environment and Consumer Protection of the Free Hanseatic City of Bremen within the ERDF programme Bremen 2014-2020 (201/PF_IWES_Zukunftskonzept_Betriebsfestigkeit_Rotorblätter_ Phase I).





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
