# Peer review of "Benefits of sub-component over full-scale blade testing elaborated on a trailing edge bond line design validation"

_Wind Energy Science, 2017_

## Short Comment (SC1) · 11 Sep 2017

This manuscript serves good argumentation about the benefits of sub-component over full-scale wind turbine rotor blade testing. Furthermore, the manuscript is very easy to follow. I have, however, some minor specific and technical comments (page/line):

1/13  testing time is . . . *shorter*

1/16  leave out *higher*; not only higher stress ratios are more realistic

2/32  on a strong floor *or a stiff wall*; sometimes the blade is pulled sideways to a stiff wall

3/2 testing *frequency*; the testing frequency is not necessarily the eigenfrequency

4/4 specimen*s*

5/F2 It is not very clear that the shown resistance envelopes represent the "worst" envelope of the whole blade from 0% to 100% blade length.

6/2 *suction*-side points towards the strong floor

6/20 $m = 10$, which is a typical value for glass fiber reinforced epoxy; add reference

9/3 *inclination* of the stress ratio distribution

10/8 Not all fibers can be considered to be isotropic, e. g. carbon fibers cannot.

10/9 Explain or give reference to what is meant by a symmetric constant life diagram (CLD)

10/9 It is also an assumption that a symmetric CLD can be used for isotropic materials. If possible, give a reference.

---

## Referee Comment (RC1) · P.A. Joosse (Referee) · 3 Nov 2017

Advise to change title to 'Benefits of sub-component over traditional full-scale ... (etc)'

---

## Referee Comment (RC2) · P.A. Joosse (Referee) · 3 Nov 2017

The paper accurately defines some drawbacks of traditional, uni-axial full-scale testing like incorrect loading and materials not loaded to the same fraction of stress exposure. However, there are some points that should be addressed to strengthen the paper.

The paper only evaluates the traditional full-scale test, while more advanced methods are merely mentioned. The statement in the paper that the advanced methods only aim at minimizing over-loading of the structure is incorrect, the aim is to have a more correct loading. An in-depth comparison of the sub-component test to advanced full-scale testing methods like bi-axial, forced response tests is advised. In this it should

be remembered that e.g. WMC in the Netherlands started with bi-axial testing already long time ago, but this was replaced by a uni-axial resonant fatigue test for practical reasons (time, money).

An inherent drawback of a sub-component test is that the selection of the critical regions and its loading and the design of the component set-up need to be done using the design models. This seems in conflict with the certification objective of the full-scale test: 'to validate the assumptions made in the design models'. How to repair this drawback?

In case a full-scale test would be replaced by sub-component tests, which seems the advice in the conclusions, how many components would then be needed to be tested? The DTU10MW example already identifies 2 regions of interest for lead-lag loading, but there will be more. Will there then still be a time benefit?

---

## Referee Comment (RC3) · Anonymous Referee #2 · 7 Nov 2017

The paper deals with the advantage of subcomponent testing of utility-scale blades over the full-scale testing. A computational study is carried out on the structurally critical regions of the blade to determine the strain levels during a full-scale test. The study provides interesting insights on strain distributions for the edge-wise and flap-wise bending modes along the length span of the blade and discusses the effect of different stress ratios on allowable fatigue cycles. Authors justify the advantage of sub-component testing over the full-scale test based on the testing duration and proximity to the target loads. The paper is well-written however a few points need to be addressed. Below is the summary of the comments:

[Figure]

-1/15 In the introduction section, the subcomponent testing is somehow presented as a substitute for full-scale testing which is not realistic. Coupon testing of the materials and the final full-scale test are both required for certification of utility-scale blades. However, subcomponent testing can bridge the gap between the coupon and full-scale tests and increase the assurance of the blade manufacturer/designer for use of new materials or designs in a blade before a full-scale test. Subcomponent testing may not replace the need for a final full-scale test but it has the potentials to be considered as a standard intermediate test for utility-scale blades. Subcomponent test can also expedite and facilitate the introduction of new materials into wind turbine blade manufacturing industry.

-4/7 Reference should be added to give the readers examples of the use of DIC technique and full-field measurements in subcomponent testing. See the papers below:

Zarouchas, D. S., Makris, A. A., Sayer, F., Van Hemelrijck, D., Van Wingerde, A. M. (2012). Investigations on the mechanical behavior of a wind rotor blade subcomponent. Composites Part B: Engineering, 43(2), 647-654.

Asl, M. E., C. Niezrecki, J. Sherwood, and P. Avitabile. "Experimental and theoretical similitude analysis for flexural bending of scaled-down laminated I-beams." Composite Structures (2017).

Asl, M. E., C. Niezrecki, J. Sherwood, and P. Avitabile. Similitude analysis of the strain field for loaded composite I-beams emulating wind turbine blades. InProceedings of the American Society for Composites: Thirty-First Technical Conference 2016 Sep.

-5/10 It might not be not clear for readers where the 10% length span is measured from. Either a figure should be added to address that or authors can explicitly mention that it's measured from the root section of the blade.

- Section 3, The paper is supposed to discuss and compare the full-scale test to subcomponent test. Although the authors' comments on full-scale testing have been fairly

supported by simulation data, there are no significant data or quantitative measures to support their comments on subcomponent testing side. Authors should either include simulation data to justify their comments on subcomponent testing or cite the references which include such data or elaborate on their reasoning to support their insights on subcomponent testing. For instance, page 8 line 11 reads "while in an SCT setup any of the loading scenarios for the different wind speed bins which are shown can theoretically be replicated..". It's not clear how this scenario could possibly be implemented in reality. There are no diagrams, figures, references or at least a detailed explanation. Same thing on page 11 lines 1 through 7.

-Authors should expand the literature review and provide the readers with a broad perspective of the different approaches and techniques that have been developed so far for subcomponent testing of wind turbine blades. In the literature review, authors should also comment on the limitations of the developed subcomponent techniques to give the readers a realistic assessment of the state-of-art techniques that have been developed to date for structural performance assessments of utility-scale wind turbine blades. This should include the experimental techniques using DIC, analytical tools such as similitude analysis and scaled subcomponents or computational models for fracture in the adhesive joints. See the papers below:

Ji, Y.M. and Han, K.S., 2014. Fracture mechanics approach for failure of adhesive joints in wind turbine blades. Renewable Energy, 65, pp.23-28.

Laustsen, S., Lund, E., Kühlmeier, L. and Thomsen, O.T., 2014. Development of a High‐fidelity Experimental Substructure Test Rig for Grid‐scored Sandwich Panels in Wind Turbine Blades. Strain, 50(2), pp.111-131.

Eydani Asl, Mohamad, et al. "Similitude analysis of thin-walled composite I-beams for subcomponent testing of wind turbine blades." Wind Engineering (2017): 0309524X17709924.

Fernandez, Garbiñe, Hodei Usabiaga, and Dirk Vandepitte. "Subcomponent development for sandwich composite wind turbine blade bonded joints analysis." Composite Structures 180 (2017): 41-62.

Asl, M. E., Niezrecki, C., Sherwood, J., Avitabile, P. (2014). Application of structural similitude theory in subcomponent testing of wind turbine blades. In Proceedings of the American Society for Composites (pp. 8-10).

---

## Author Comment (AC1) · 5 Dec 2017

Thank you very much for your feedback. In the following we would like to react on your comments in detail.

1/13  testing time is . . . *shorter*

  R  Rectified as proposed.

1/16  leave out *higher*; not only higher stress ratios are more realistic

  R  Rectified as proposed

2/32 on a strong floor *or a stiff wall*; sometimes the blade is pulled sideways to a stiff wall

R  Rectified as proposed.

3/2 testing *frequency*; the testing frequency is not necessarily the eigenfrequency

R  Rectified as proposed.

4/4 specimen*s*

R  Rectified as proposed.

5/F2 It is not very clear that the shown resistance envelopes represent the "worst" envelope of the whole blade from 0% to 100% blade length.

R  The caption of F2 has been emphasized.

6/2 *suction*-side points towards the strong floor

R  Rectified as proposed.

6/20 $m = 10$, which is a typical value for glass fiber reinforced epoxy; add reference

R  Rectified as proposed.

9/3 *inclination* of the stress ratio distribution

R  Rectified as proposed.

10/8 Not all fibers can be considered to be isotropic, e. g. carbon fibers

R  Rectified as proposed.

10/9 Explain or give reference to what is meant by a symmetric constant life diagram (CLD)

R Reference was added.

10/9 It is also an assumption that a symmetric CLD can be used for isotropic materials. If possible, give a reference.

R Reference was added.

Please also note the supplement to this comment:
https://www.wind-energ-sci-discuss.net/wes-2017-35/wes-2017-35-AC1-supplement.pdf

[Figure]

**Supplement:**

[revised manuscript text omitted]

---

## Author Comment (AC2) · 8 Dec 2017

Thank you very much for taking the time to review our manuscript and giving us your feedback. In the following, please, find our responses (R) to your comments (C):

C "The paper only evaluates the traditional full-scale test, while more advanced methods are merely mentioned. The statement in the paper that the advanced methods only aim at minimizing over-loading of the structure is incorrect, the aim is to have a more correct loading."

R The respective part in the introduction was rectified.

[Figure]

C "An in-depth comparison of the sub-component test to advanced full- scale testing methods like bi-axial, forced response tests is advised. In this it should be remembered that e.g. WMC in the Netherlands started with bi-axial testing already long time ago, but this was replaced by a uni-axial resonant fatigue test for practical reasons (time, money)."

R We have added the comparison to resonant bi-axial blade testing to the paper. According to your advice we have added a sentence on forced response testing.

C "An inherent drawback of a sub-component test is that the selection of the critical regions and its loading and the design of the component set-up need to be done using the design models. This seems in conflict with the certification objective of the full-scale test: 'to validate the assumptions made in the design models'. How to repair this drawback? "

R FST also underlies the described drawback. The selection of critical regions is also done using the design model. The positions of load frames, for example, is decided based on the critical regions in the static test. The general idea behind the model validation by experiment is to bring your model iteratively closer to the boundaries of the experiment. At some point you need to predict the behavior of the specimen, the only thing you can do is to trust your model.

C "In case a full-scale test would be replaced by sub-component tests, which seems the advice in the conclusions, how many components would then be needed to be tested? The DTU10MW example already identifies 2 regions of interest for lead-lag loading, but there will be more. Will there then still be a time benefit?"

R The intention of this paper is not to give arguments that full-scale blade testing should be replaced by sub-component testing. The idea is to have a further intermediate level experimental method at hand to validate models under more realistic loading conditions. To eliminate misunderstandings, we have added a

sentence to the introduction and some application case scenarios to the conclusions.

Please also note the supplement to this comment:
https://www.wind-energ-sci-discuss.net/wes-2017-35/wes-2017-35-AC2-supplement.pdf

---

## Author Comment (AC3) · 8 Dec 2017

Thank you very much for taking the time to review our manuscript and giving us your feedback. In the following, please, find our responses (R) to your comments (C):

C "1/15 In the introduction section, the subcomponent testing is somehow presented as a substitute for full-scale testing which is not realistic. Coupon testing of the materials and the final full-scale test are both required for certification of utility-scale blades. However, subcomponent testing can bridge the gap between the coupon and full-scale tests and increase the assurance of the blade manu-

facturer/designer for use of new materials or designs in a blade before a full-scale test. Subcomponent testing may not replace the need for a final full-scale test but it has the potentials to be considered as a standard intermediate test for utility-scale blades. Subcomponent test can also expedite and facilitate the introduction of new materials into wind turbine blade manufacturing industry."

R The intention of this paper is not to give arguments that full-scale blade testing should be replaced by sub-component testing. The idea is to have a further intermediate level experimental method at hand to validate models under more realistic loading conditions. To eliminate misunderstandings, we have added a sentence to the introduction and some application case scenarios to the conclusions.

C "4/7 Reference should be added to give the readers examples of the use of DIC technique and full-field measurements in subcomponent testing. See the papers below:

Zarouchas, D. S., Makris, A. A., Sayer, F., Van Hemelrijck, D., Van Wingerde, A. M. (2012). Investigations on the mechanical behavior of a wind rotor blade subcomponent. Composites Part B: Engineering, 43(2), 647-654.

Asl, M. E., C. Niezrecki, J. Sherwood, and P. Avitabile. "Experimental and theoretical similitude analysis for flexural bending of scaled-down laminated I-beams." Composite Structures (2017).

Asl, M. E., C. Niezrecki, J. Sherwood, and P. Avitabile. Similitude analysis of the strain field for loaded composite I-beams emulating wind turbine blades. In Proceedings of the American Society for Composites: Thirty-First Technical Conference 2016 Sep."

C "Authors should expand the literature review and provide the readers with a broad perspective of the different approaches and techniques that have been developed so far for subcomponent testing of wind turbine blades. In the literature

review, authors should also comment on the limitations of the developed subcomponent techniques to give the readers a realistic assessment of the state-of-art techniques that have been developed to date for structural performance assessments of utility-scale wind turbine blades. This should include the experimental techniques using DIC, analytical tools such as similitude analysis and scaled subcomponents or computational models for fracture in the adhesive joints. See the papers below:

Ji, Y.M. and Han, K.S., 2014. Fracture mechanics approach for failure of adhesive joints in wind turbine blades. Renewable Energy, 65, pp.23-28.

Laustsen, S., Lund, E., Kühlmeier, L. and Thomsen, O.T., 2014. Development of a High fidelity Experimental Substructure Test Rig for Grid scored Sandwich Panels in Wind Turbine Blades. Strain, 50(2), pp.111-131.

Eydani Asl, Mohamad, et al. "Similitude analysis of thin-walled composite I-beams for subcomponent testing of wind turbine blades." Wind Engineering (2017): 0309524X17709924.

Fernandez, Garbiñe, Hodei Usabiaga, and Dirk Vandepitte. "Subcomponent develop- ment for sandwich composite wind turbine blade bonded joints analysis." Composite Structures 180 (2017): 41-62.

Asl, M. E., Niezrecki, C., Sherwood, J., Avitabile, P. (2014). Application of structural similitude theory in subcomponent testing of wind turbine blades. In Proceedings of the American Society for Composites (pp. 8-10)."

R The proposed references deal with blade elments and details. According to DNVGL guideline 2015 we consider a rotor blade substructure as a blade subcomponent. Therefore, we have added a reference to DIC w.r.t. full-scale blade testing. Furthermore, we have taken out the reference dealing with element/detail testing by Sayer et al. (2012).

C "5/10 It might not be not clear for readers where the 10from. Either a figure should be added to address that or authors can explicitly mention that it's measured from the root section of the blade."

R The definition was added.

C "Section 3, The paper is supposed to discuss and compare the full-scale test to sub- component test. Although the authors' comments on full-scale testing have been fairly supported by simulation data, there are no significant data or quantitative measures to support their comments on subcomponent testing side. Authors should either include simulation data to justify their comments on subcomponent testing or cite the references which include such data or elaborate on their reasoning to support their insights on subcomponent testing. For instance, page 8 line 11 reads "while in an SCT setup any of the loading scenarios for the different wind speed bins which are shown can theoretically be replicated..". It's not clear how this scenario could possibly be implemented in reality. There are no diagrams, figures, references or at least a detailed explanation. Same thing on page 11 lines 1 through 7."

R Rosemeier et al. (2016) is referenced wherein the SCT principle is described and shown that FST conditions can be reached. Additionally, we give the advice that it is possible to replicate arbitrary loading conditions with the described method through shifting the ball joint positions within the cross-section. We think that the references and explanations given are sufficient to follow our conclusions.

Please also note the supplement to this comment:
https://www.wind-energ-sci-discuss.net/wes-2017-35/wes-2017-35-AC3-supplement.pdf

**Supplement:**

[revised manuscript text omitted]

---

## Author Comment (AC4) · 8 Dec 2017

It is not necessary to change the title since the study has been extended to cover all relevant FFST scenarios.
* * *

---

## Editor Comment (EC1) · J.ÂăN. Sørensen (Editor) · 16 Dec 2017

I think that the authors have thoroughly responded on the reviewer comments, which suggested minor corrections. If the two reviewers now are pleased with the answers from the authors, I suggest the paper be accepted with the proposed modifications. Reviewers: Can you please respond back, if you deem the paper to be accepted for publication or if you want further corrections?

---

## Referee Comment (RC4) · Anonymous Referee #2 · 25 Dec 2017

The authors addressed some of the comments fairly, however, a few comments need to be addressed effectively.

The second and third comments about the literature review on sub-component testing should be addressed. The title of the paper is on sub-component testing but the paper lacks a rigorous literature review on this subject. The authors should improve the literature review so that a wide spectrum of the readers, specially the ones who are less familiar with the novel concept of the sub-component testing get engaged. Although the authors discussed a very specific type of subcomponent in their study, they should still include an exhaustive literature review and discuss the other methods that have

been developed so far and elaborate on their advantages and shortcomings (at least look at the works by JF Mandell, F. Sayer, D. Zarouchas, ME Asl and G fernandez).

The fifth comment (supporting of the finding for sub-component testing) has not been addressed effectively. The authors have discussed the FFST and its respective stress ratios and testing time in detail, however, the SCT part lacks a rigorous discussion.

---

## Author Comment (AC5) · 10 Jan 2018

Thank you very much for taking the time to review our manuscript and giving us your feedback. In the following, please, find our responses (R) to your comments (C):

C "The second and third comments about the literature review on sub-component testing should be addressed. The title of the paper is on sub-component testing but the paper lacks a rigorous literature review on this subject. The authors should improve the literature review so that a wide spectrum of the readers, specially the ones who are less familiar with the novel concept of the sub-component

testing get engaged. Although the authors discussed a very specific type of sub-component in their study, they should still include an exhaustive literature review and discuss the other methods that have been developed so far and elaborate on their advantages and shortcomings (at least look at the works by JF Mandell, F. Sayer, D. Zarouchas, ME Asl and G fernandez)."

R As already mentioned in the previous comment (AC3), the proposed references deal with generic elements and details, such as beam specimens, and not with blade substructures. According to DNVGL guideline 2015, we consider a rotor blade substructure as a full scale blade sub-component, which is a cut-out part of the real blade. Since the whole work deals with sub-components in the sense of cut-out blade parts, we see no necessity to mislead the reader with a literature review on element and detail testing. To the authors' knowledge, the corresponding available literature is limited. To further highlight the means of SCT, we have added a further reference by Kühlmeier (2006). The intention of this paper is not to give a review of all levels of the testing pyramid but rather focus on the comparison of the top of the pyramid (FST) with the next lower level (SCT).

C "The fifth comment (supporting of the finding for sub-component testing) has not been addressed effectively. The authors have discussed the FFST and its respective stress ratios and testing time in detail, however, the SCT part lacks a rigorous discussion."

R As already mentioned in the previous comment (AC3), simulation results on SCT compared to FST, as well as the general idea of the SCT concept are shown in reference Rosemeier et al. (2016). Based on the findings in Rosemeier et al. (2016) it is assumed that the SCT concept is applicable to replicate any load direction vector. Furthermore, the implementation of such testing scenarios is highlighted on p. 3, l. 9: "The position of the two joints within the cross-sectional plane can be chosen arbitrarily, which makes it possible to introduce any load

combination and distribution of lead-lag and flap-wise loading." For clarification, the idea of the testing scenario was repeated on p. 10, l. 6 (diff_06_04.pdf) and referenced again.

Please also note the supplement to this comment: https://www.wind-energ-sci-discuss.net/wes-2017-35/wes-2017-35-AC5-supplement.pdf

---

## Author Comment (AC6) · 12 Jan 2018

Erroneously, we have given DNVGL guideline 2015 as reference for the definition of a "sub-component" or an "element/ detail". Correctly, the terms that we refer to are defined in the upcoming standard IEC61400-5 revU WIND ENERGY GENERATION SYSTEMS – Part 5: Wind turbine blades (currently open for comments). Since this standard is not yet publicly available, the relevant excerpt is shown in the following:

**6.1.2 Building Block Approach for Composite Structural Design**

The traditional detailed design (analytic and numerical calculation together with validated material data and full blade testing) of FRP structures can be enhanced by a building-block approach, starting with coupon-level tests, analysis and testing of more complicated structures; and culminating in a full blade test. This relationship is shown in Figure 1, where increasingly more complex tests are developed to evaluate more complicated loading conditions and failure modes.

The approach can be summarized as follows:

**Coupons:** A number of tests are conducted at the coupon level, where confidence in repeatable physical properties is developed. Procurement specifications are developed for the individual constituents, and allowable design variables developed for lamina/laminate combinations.

**Elements and Details:** Critical areas from the design analysis identify elements for further testing and analysis at the design conditions with representative specimens. This may include such tests as the spar cap to web bond line or ply drops in the spar cap laminate.

**Sub-Components:** Parts and sections representative of the blade design are tested to evaluate specific loading conditions and failure modes. Examples include spars, shells and root sections. The test components may be full or partial scale where demonstrated to be representative.

**Full Blade:** A full blade or significant part of a blade, representative of the blade design is tested to evaluate specific loading conditions and failure modes. The blade may be full or partial scale where demonstrated to be representative.

The number of tests required for each level should be tailored for each design activity, with the blade designer responsible for the development of a reasonable number of

tests at each stage.

Tests on the element and detail as well as sub-component level will increase the confidence in the structural design.

For design values (strength, stiffness, etc.) developed from test at any building block level (material sample, sub-component, etc.), the validity of such design values shall be described and limited by acceptance criteria and tolerances to be met in the final design.

Please also note the supplement to this comment:
https://www.wind-energ-sci-discuss.net/wes-2017-35/wes-2017-35-AC6-supplement.pdf
* * *
[Figure]

[Figure]

**Fig. 1.**

**Supplement:**

[revised manuscript text omitted]